# The Role of Sleep Quality in the Psychological Well-Being of Final Year UndergraduateStudents in China

**DOI:** 10.3390/ijerph15122881

**Published:** 2018-12-15

**Authors:** Keyu Zhai, Xing Gao, Geng Wang

**Affiliations:** 1School of Education, University of Glasgow, Glasgow G12 8QQ, UK; k.zhai.1@research.gla.ac.uk (K.Z.); g.wang.3@research.gla.ac.uk (G.W.); 2The Bartlett School of Planning, University College London, London WC1E 6BT, UK

**Keywords:** sleep quality, psychological well-being, Chinese final year university students, higher education, gender

## Abstract

There are increasing numbers of university students in China suffering from poor sleep and psychological well-being problems. In particular, the issues are more severe among the final year undergraduate students, because they are experiencing a transitory period from university life to the workplace. However, extant research has rarely explored sleep quality and psychological well-being of final year university students. To better understand the role of sleep quality in psychological well-being, we examined the association between different sleep quality and mental health. Based on a cross-sectional survey of 2495 full-time final year university students in China, we employed multivariable logistic regression to assess association between sleep quality and psychological well-being by controlling for sociodemographic factors such as age, gender, education, marital or relationship status, household conditions, place of birth, study subjects and etc. According to the research results, we can find strong association between sleep quality and psychological well-being. Having normal sleep quality is associated with lower level of psychological well-being problems. By contrast, poor sleep quality is associated with high level of negative psychological well-being. Poor sleep quality has higher potency than normal sleep quality due to negative bias. Among covariates, age, gender and education have significant effects on psychological well-being.

## 1. Introduction

Psychological well-being of university students is an important research field, and thus this issue increasingly receives global concern [1]. Taking the UK as an example, the UK Psychiatric Morbidity Survey reported significant increases in anxiety and depression among the young people aged from 16 to 24 [1]. Rich research identifies positive mental health serves as a protective role against health risk behaviors, but negative mental health would result in risky behaviors among university students [2]. In addition, a large number of cross-sectional evidence shows that individual having much depression and hopeless tends to be less physically active [3] and has more negative feelings, such as suicidal thoughts [4]. Macaskill [1] studied the students from academic year for first, second and third year in the UK, and found rates of mental illness in students are same as that of the general population, whereas only 5.1% of them can receive treatment. Therefore, easier access to specialist treatment is in great need among university students. By contrast, some extant research finds that Hong Kong university students are more vulnerable than their peers regarding mental disorders, because their health is poorer and they have higher rates of getting depression and anxiety [5,6]. University students are five times more likely to be recognized with mental health issues [7]. The different context determines the different research results, and thus studying university students’ sleeping quality and psychological well-being should be closely related to the specific context. There are a variety of factors influencing university students’ mental health. For university students, their failure to meet basic physical activity guidelines [8] and increasing sedentary behaviors [9] are significant factors. Among a wide variety of variables, sleep quality is a significant factor [10], and sleep disorder frequently co-occurs with mental health problems.

Poor sleep quality is a symptom, and it is featured by difficulty of falling and remaining asleep [11,12]. Extensive research has shown that sleep quality has become an increasing public health focus, and poor sleep quality leads to increased risk of mental problems [13], including depression and anxiety [14]. Nowadays, poor sleep quality is a prevalent symptom among young adults, affecting over 10% of the adult population [15]. Among these young adults, there are plenty of university students [16]. Voelker [17] discussed that university students’ biological sleep rhythms are disrupted, because they experience more stress regarding their futures and employment or late-night computer work. Besides, academic work, later bedtime resulted from social integration demand, environmental noise are disturbing undergraduate students’ sleep quality [18,19]. In addition, a growing body of research has documented the gendered difference in mental health [20,21]. Therefore, among university students, it can be concluded that poor sleep quality is a common problem and has increased risk for their mental health [22]. In order to discover the role of sleep quality in mental health, much research has been conducted, but most of extant literature focuses on specific group such as the elderly or young adults. There is few research concentrating in studying university students [22,23,24]. Besides, final year university students in China are a special group, because they are undergoing a critical transitory period in which they are going from university life to adulthood and making major life decisions [25]. More importantly, Suen et al. [26] found increasing academic year in university is associated with increased rate of getting sleep problems. Last but not least, the Chinese context is very different from typical Western education systems. The common accommodation of undergraduate student in China is a special issue. In China’s student accommodation, several students share a same bedroom. The relationship with roommates becomes an important factor, and it make the research of sleep problem more complicated. Facing the research background, this paper addresses what significant factors affect sleep quality, and how poor sleep quality affects psychological well-being of final year Chinese undergraduate students.

Motivated by the research question, this paper aims to examine, in a large cohort of university students, associations between sleep quality and psychological well-being of final year university students in China. We would investigate how sleep quality affects psychological well-being and compare the effects sizes associated with positive and negative quality on psychological well-being in the same statistical model.

This paper contributes to bridge two gaps. First, this paper initially explores the role of sleep quality in psychological well-being, targeting final year Chinese university students. China’s is very different from the Western education system. China has the largest number of undergraduate students and the special shared student accommodation, and the fast social changes and high level of social competition dramatically influence young adults in every aspects [27]. Facing the fact that a large proportion of Chinese university students are suffering from poor sleep quality [28] and mental health problems [2], this paper is timely and necessary. Another gap that would be filled is to provide overall analysis of psychological well-being of Chinese university students. This paper would analyze a series of factors of final year university students’ psychological well-being. 

## 2. Method

The study conducted a cross-sectional data of sleep quality and psychological well-being among final year undergraduate students from large China’s university. In March 2018, participants were recruited by WeChat. We invited the participants to finish an online survey to understand the effect of sleep quality on mental health. According to power calculations which depend on the prior distributions and estimates of our variables [29], our survey demanded at least 2600 participants. To be eligible, the participants must be final year undergraduate students and were expected to obtain a Bachelor degree in July 2018. There were totally 2879 eligible participants. Finally, 2495 participants provided complete results for our survey. This study was approved by the Ethics Committee of College of Social Science of University of Glasgow on 1st September 2017 (No. 400160231).

Participants provided online informed consent at the beginning of the questionnaire, and the mean survey completion time was 12.8 min. After participants finished the survey, the WeChat system would randomly generate a voucher ranging from 1 to 10 Yuan to thank them. Additionally, this could attract more participants and to some degree increase the questionnaire completion quality. Because the voucher was sent out by the WeChat software, the data was still anonymized. Collecting a voucher in WeChat does not need the participants’ personal information. This study was funded by a Doctoral Fellowship from the China Scholarship Council.

### 2.1. Measure of Psychological Well-Being

In this study, we chose the K6 scale of mental well-being to measure the mental disorders, classified according to the Diagnostic and Statistical Manual of Mental Disorders, 4th Edition (DSM-IV). The K6 is the reliable and valid instrument to screen for any DSM-IV disorder [30], and additionally this measurement has been widely used to investigate the mental health of different population in China [31,32,33]. The K6 scale designs diagnose cases with non-specific mental illness in the general population, and is applied in national survey which is carried out by World Mental Health Initiative of World Health Organization. Regarding the K6 scale, our study took advantage of five items, including nervous, anxious, worthless, hopeless and depressed. We asked respondents how often do you feel the above five kinds of emotion within a past month. The results can range from 1 (often) to 4 (never). Thus, higher scores mean better mental health. 

The results of psychological well-being were viewed as the dependent variable in the study. Due to the non-normal distribution and skewed of mental health, it cannot be served as a continuous variable [29]. Therefore, following Primack et al. [29], we processed the data of psychological well-being in the following ways to improve interpretability. Standard cutoff for depression of 11, corresponding to T-score with 60.5 [34], is first used to make the mental health based on the recommendation from American Psychiatric Association [1]. Second, because the K6 scale is for evaluating the levels of mental health, the study divided the results into three equal types—high (20); medium (17–19); low (5–16). In the first analysis, standard cutoff was applied, and three level types were used in our secondary analyses.

### 2.2. Sleep Quality

Pittsburgh Sleep Quality Index (PSQI) has been validated for application in different types of population samples [35,36]. Sleep quality in the last month was assessed via three pre-specified response categories: poor quality, impaired quality and normal quality. Guo et al. [36] indicated that poor quality means that participants have sleep deprivation symptoms, like taking sleeping peels, or insomnia; participants with normal sleep quality have the following characteristics: difficult to weak up at night, enjoying a high mood in daytime, and falling sleep within 30 min; impaired quality refers to the sleep status between poor and normal sleeping quality. However, in the process of survey, participants find it difficult to define their appropriate categories. Thus, based on the above discussions, the study divided sleep quality into two categories: normal quality and poor quality. We evaluated normal and poor sleep quality through participants’ self-rated in the questionnaire. Our survey provided sliders between 0 and 100 for participants. Participants assessed the percentage of their sleep quality involved normal and poor status respectively. According to Primack et al. [29], in order to conduct logistic regression and improve results’ interpretability, the study changed the participants’ responses to a 10-points scale, and 1 point is equal to 10%. This transformation is based on responses’ natural distribution.

### 2.3. Covariates

We controlled for sociodemographic factors as our covariates that are known to be related to psychological well-being and sleep quality, including education [37], household conditions [38], marital or relationship status [39], physical activity [40], and illness health [41]. In our study, education is based on the official administrative tier of China’s universities, and higher level of universities means better educational quality, which may be positively related to mental health. Household conditions focus on socioeconomic status of parents. We employed Lu’s division standard, which is widely used in research of socioeconomic status [42]. Regarding physical activity, following Mai et al. [43], we divided it into three types: physically inactive, less active (meaning 1–4 occasions conducted exercise or sport within a past month), and more active (meaning 5 or more). We also investigated Body Mass Index (BMI) which is derived from weight and height to measure illness health conditions [43], and the study placed it into physical activity. 

In addition to above covariates, the study also asked participants to show their place of birth, subjects, gender and age. Age was treated as a continuous variable in years, ranging from 18 to 30 [29]. Place of birth was collapsed into two categories: rural and urban. Cheung [31] indicated that students from rural areas are more likely to suffer from psychological well-being problems. In our study, we were interested in questioning whether sleep quality of final year students from different subjects is different in mental health. Thus, we also investigated the subject information of participants. 

### 2.4. Analysis Strategy

The study summarized the sample through depicting counts and percentage. According to the variables, in addition to continuous variables, there are plenty of sociodemographic variables. Multivariable models are usually preferentially selected when there are categorical variables with many categories [44]. Multivariable model has been commonly used in biometrical applications to model the associations between a result and predictor variables [44,45]. We also performed *t*-tests to evaluate the sample mean differences on continuous, and Chi-square tests for the categorical variables. The significant values of all tests were presented. In order to meet the assumptions of analytic models, the study controlled for collinearity among all covariates. The largest Pearson’s correlation coefficient was 0.598, indicating that multicollinearity issues among independent variables were not a challenge. Nonsignificant *p*-values (from 0.93 to 0.98) indicated that the models using ordered logistic regression meet the proportional odds assumption [29].

Then, by employing logistic regression and ordered logistic regression, the study evaluated the effects of sleep quality on mental health. In multivariable models, the study chose a priori to adjust for all covariates. The statistical significance of a two-tailed test (α = 0.05) was defined. Using Stata 14, we analyzed all data. Based on the method of Primack et al. [29], there were three steps of sensitivity analyses to explore results’ robustness. We first used continuous variables to conduct analysis; second, extant research shows a *p* < 0.20 can avoid the overcontrolling [29]. And thus the study analyzed data by using covariates with a *p* < 0.20; third, in order to make sure that the results are consistent with the first model, we conducted the analysis by taking advantage of psychological well-being cutoffs. Due to similar results between primary models and additional analysis, this paper only demonstrated the primary results and analysis.

## 3. Results

Without statistically taking any confounders into account, Table 1 reports the descriptive statistics of variables. *t*-Tests and Chi-square were used to evaluate mean differences of continuous variables and the categorical variables, respectively. They indicated the variance inflation factor. The reported psychological well-being is significant. As expected, the most of final year students suffer from poor sleep quality. As shown in Table 1, gender distribution was relatively balanced. The distribution of educational background was also in line with the administrative level of Chinese universities. Household conditions were based on the father’s occupation.

There were 37% of students being in love or married. 37% of participants were physically inactive, whereas 14% were more active. The mean (26.51) of BMI belongs to overweight. 39% of participants lived in rural areas, and 61% were from urban areas. In terms of subjects, the majority of participants (42%) were studying Engineering, and 35% and 23% studied Science and Humanities and Social Sciences, respectively. The age of our participant ranged from 18 to 30 years old, whose mean was 22.7 (SD = 3.2).

When psychological well-being was operationalized in tertiles, variables, including normal sleep quality, poor sleep quality and gender, show the significance relation to mental health (Table 2). Internal consistency of the four psychological well-being items is high (α=0.90). According to data, we found they were skewed to the right, with a mean value of 7.56 (SD = 3.6) and median of 7.1 (IQR = 5–20) on the composite scale from 5 to 20. Around 35% of sample was in the low level of mental health psychological well-being while a similar percentage was in the high level category (30%). Moreover, the distribution of all independent variables was skewed. About 11% (SD = 19) of sample reported that their sleep was normal, with a median of 63% (IQR = 27–86). Over 78% of participants reported that their sleep quality is poor, with a median of 72% (IQR = 30–95).

The first group of multivariable models (see Table 3) explored relations with psychological well-being as a dichotomous variable. Model 1 includes normal sleep quality and the Model 2 includes poor sleep quality. Model 3 includes both of these independent variables. All relevant covariates are included in the Table 3. In models, AOR refers to adjusted odds ratio. CI represents confidence interval. In Model 1, each 10% increase in normal sleep quality would result in a 3% decrease in psychological well-being problem (AOR = 0.97; 95% CI = 0.92–1.003). In Model 2, each 10% increase in poor sleep quality was associated with a 27% increase in psychological well-being problem (AO = 1.27; 95% CI = 1.12–1.32). In Model 3, both these independent variables were in the same model, and the results were very similar. In Model 3, each 10% increase in normal sleep quality contributed to a 5% decrease in psychological well-being problem while each 10% increase in poor sleep quality was associated with 26% increase in psychological well-being problem. In the three models, covariates, including gender (male), education (211 project university) and physical activity (body mass index), significantly associated with negative mental health.

The second group of multivariable models explored associations with psychological well-being operationalized in tertiles. There are three models in the Table 4. Model 1 includes normal sleep quality, Model 2 includes poor sleep quality and Model 3 includes both of these independent variables. It had similar results compared with the first set of multivariable models. Model 1 in the Table 4, each 10% increase in normal sleep quality resulted in 4% decrease in negative mental health, which was of significance in statistics (AOR = 0.96; 95% CI = 0.91–0.99). In Model 2, each 10% increase in poor sleep quality contributed to a 21% increase in negative mental health. Model 3 reported the similar results compared with Model 1 and 2. In three models, gender (male), household conditions and education had significant relation with mental health. In Model 1, subject (engineering) was significantly associated with mental health. In Model 2, age (25–30) significant associated with mental health.

## 4. Discussion

This study of final year undergraduate students in China found that normal sleep quality was weakly associated with lower level of negative psychological well-being condition, but having poor sleep quality had strong association with higher level of negative mental health. The sample in this study was within age ranging from 18 to 30 years old, covering the most conditions of final year university students in China, so this study is valuable to know the associations between their sleep quality and mental health. These findings may persuade final year university students to pay attention to their sleep quality. Because final year university students in China are undergoing transitions from university to society, they have more mental disorders than their peers [25]. University, especially in the final year, is an important transient stage in which there is a variety of pressure such as academic, financial and interpersonal pressures.

One potential explanation for the effects of sleep quality on psychological well-being is that individual sleep quality is influenced by various factors in different ways, including socioeconomic factors [46], academic, financial and interpersonal pressures [25], and transitory challenges [47]. The optimistic and pessimistic perceptions of future development is different among the final year university students. Because different participants have different situations, some participants tend to have normal sleep quality based on minor pressure and promising career development while some others engage in much pressure and less support and then show poor sleep quality that may result in mental problems.

The research findings reconfirm the prevalence of poor sleep quality in university [22] and the associations between sleep quality and psychological well-being [28]. Normal sleep quality is an important physiological issue for human beings, but poor sleep quality would bring serious psychological well-being problem [48]. Depression and anxiety will appear when sleep quality becomes poor. In our model including both independent variables, each 10% increase in normal sleep quality contributed to a 5% decrease in mental problem. The result was statistically significant, and it is also consistent with studies reporting the important role of sleep quality in mental health. However, the magnitude of the results were much stronger regarding poor sleep quality. Each 10% increase in poor sleep quality would result in a 26% increase in mental problem. It is interesting that the poor sleep quality has larger effect size than normal sleep quality. One may conclude that poor sleep quality may be more potent than normal sleep quality. This reasoning conforms to negativity bias concept. This concept reports a tendency of human beings to give more emphasis to negative entities compared with positive ones [49].

In addition to normal sleep quality and poor sleep quality, there are three covariates associated with mental health. Gender, university type and age are three significant factors, and they relate mechanically each other regarding university student’s sleep quality and mental health. First, gendered issue is the most significant one. Male participants had approximately 1.5-fold odds of increased mental problem than female participants. This result is divergent from plenty of extant research. Existing research reports the female university students have higher proportion of poor sleep quality [28,50,51]. It may result from China’s social pressure of male university students, resulted from China’s actual conditions. In Chinese society, the male has more opportunity of getting higher education and higher income than the female, but the male has lower happiness, caused by high expectation for male and huge pressure designed for Chinese male [52]. Second, participants from 211 project universities had 85% odds of increased mental problem compared with those in 985 project universities. Although it is consistent with the high level of academic pressure [25], it could also result from the fierce competition among 211 project universities. Third, participants within age ranging from 25 to 30 have 15% increase in mental problem than 18-year-old participants, which shows the increasing age is significantly associated with mental health.

Although the early findings need to be replicated, they might be useful to university students’ psychological well-being practitioners. Facing the special group, the final year undergraduate students in China, specific strategies are required to improve sleep quality. In China, undergraduate students have common accommodation in which several students share one bedroom, the pressure of interpersonal communication should be paid attention to.

## 5. Conclusions

This study initially explores the association between sleep quality and psychological well-being among the final year undergraduate students in China. We found that having normal sleep quality is associated with lower level of mental problem. By contrast, poor sleep quality is associated with high level of negative mental health. The magnitude of effect seems to be more substantial and significant for participants having poor sleep quality. Therefore, it could be helpful to increase awareness of the importance of avoiding poor sleep quality of final year undergraduate students in China so that we can reduce the risk of mental problem. The major limitation of this study is the nature of cross-sectional data. For future study, using longitudinal data would be useful to examine the directionality of results. Additionally, we employed K6 scale to assess mental health. The measurement of sleep quality and psychological well-being is assessed by a questionnaire. Although the sample is large enough to analyze, it is a self-report scale. Instead, it is not the gold standard that involves interview under a psychiatric professional. Lastly, the lack of sleep duration metrics is also another limitation. It is very hard to measure how many sleeping hours a participant have. However, it is part of measurement of sleep quality. In future research, techniques can be used to compute sleep duration.

## Figures and Tables

**Table 1 ijerph-15-02881-t001:** Descriptive statistics (*N* = 2495).

Variables	Mean/Sample %	(SD)	Test Statistics ^a^	*p* ^b^
Psychological well-being	14.38	(4.02)	*t* = 1.22	*p* = 0.004
Normal sleep quality	49	(19.07)	*t* = 0.56	*p* = 0.001
Poor sleep quality	78	(32.13)	*t* = 0.41	*p* = 0.002
Gender				
Female	47%		*X*^2^ = 4.313	*p* = 0.036
Male	53%			
Education				
985 project	11%		*X*^2^ = 1.66	*p* = 0.017
211 project	17%			
Yi Ben	19%			
Er Ben	28%			
San Ben	25%			
Household conditions				
State & society administrator (cadre in governments), Manager (Manager or department leader in a state company, private company, foreign company and joint company)	7%		*X*^2^ = 4.69	*p* = 0.023
Private enterprise owner (having large scale company), Technological specialist (teacher, engineer, doctor, academic researcher, lawyer, culture worker)	15%			
Clerk & office worker (secretary, accountant, computer operator), Individual entrepreneur (one-person company with unlimited liability)	29%			
Commercial & service worker (tertiary industry workers), Industrial worker (worker in factory, mining and transport industry)	35%			
Farmer, Unemployed	14%			
Marital or relationship status				
Yes	37%		*X*^2^ = 3.03	*p* = 0.002
No	63%			
Physical activity				
Physically inactive	37%		*X*^2^ = 5.31	*p* = 0.000
Less active (1–4 occasions per month)	49%			
More active (5+ occasions per month)	14%			
Body mass index; mean (SD)	26.51	(4.7)	*t* = 3.08	*p* = 0.016
Place of birth				
Rural	39%		*X*^2^ = 1.75	*p* = 0.015
Urban	61%			
Subjects				
Humanities and Social Sciences	23%		*X*^2^ = 1.75	*p* = 0.046
Science	35%			
Engineering	42%			
Age			*t* = 4.43	*p* = 0.033
18	2%			
19–20	3%			
21–24	85%			
25–30	10%			

^a^ Test-statistics refers to *t*-test, assessing the two-sample mean differences of variables. Chi-tests is for assessing the categorical variables, including gender, education household conditions and household conditions. About undergraduate education, five tiers of universities were used, including 985 projects university, 211 project university, Yiben, Erben and Sanben. The tier is the basic university classification in China. ^b^ The *p* means the *p*-value, representing the significance in statistics.

**Table 2 ijerph-15-02881-t002:** Whole sample characteristics & bivariable associations (*N* = 2495).

Variables	Psychological Well-Being			*p* ^b^
	High (20)	Medium (17–19)	Low (5–16)	
Normal sleep quality, mean (SD)	52 (33.07)	50 (29.45)	43 (35.51)	0.000
Normal sleep quality, median (IQR ^a^)	71 (30, 88)	66 (27, 87)	59 (26, 85)	0.002
Poor sleep quality, mean (SD)	77 (31)	75 (30)	80 (33)	0.001
Poor sleep quality, median (IQR)	71 (26, 97)	73 (25, 98)	74 (31, 99)	0.005
Gender, n (%) ^c^				0.001
Female	347 (63)	743 (74)	812 (86)	
Male	207 (37)	259 (26)	127 (14)	
Education				0.015
985 project	121 (15)	147 (16)	115 (16)	
211 project	149 (18)	166 (18)	145 (20)	
Yi Ben	153 (19)	179 (19)	139 (19)	
Er Ben	208 (25)	224 (24)	193 (26)	
San Ben	196 (23)	219 (23)	141 (19)	
Household conditions				0.004
State & society administrator (cadre in governments), Manager (Manager or department leader in state company, private company, foreign company and joint company)	103 (16)	122 (13)	98 (13)	
Private enterprise owner (having large scale company), Technological specialist (teacher, engineer, doctor, academic researcher, lawyer, culture worker)	135 (17)	183 (19)	127 (17)	
Clerk & office worker (secretary, accountant, computer operator), Individual entrepreneur (one-person company with unlimited liability)	192 (24)	213 (22)	198 (27)	
Commercial & service worker (tertiary industry workers), Industrial worker (worker in factory, mining and transport industry)	237 (30)	269 (28)	201 (27)	
Farmer, Unemployed	133 (17)	166 (17)	118 (16)	
Marital or love status				0.016
Yes	293 (42)	377 (30)	193 (35)	
No	398 (58)	879 (70)	355 (65)	
Physical activity				0.001
Physically inactive	273 (43)	476 (36)	231 (43)	
Less active (1–4 occasions per month)	309 (49)	632 (48)	287 (53)	
More active (5+ occasions per month)	49 (8)	219 (16)	19 (4)	
Body mass index	665 (27)	1579 (63)	251 (10)	0.006
Place of birth				0.003
Rural	388 (47)	428 (38)	233 (43)	
Urban	436 (53)	698 (62)	312 (57)	
Subjects				0.020
Humanities and Social Sciences	186 (22)	253 (25)	122 (19)	
Science	252 (30)	319 (31)	165 (26)	
Engineering	403 (48)	458 (44)	339 (54)	
Age, year, n (%)				0.017
18	94 (14)	153 (12)	84 (14)	
19–20	116 (17)	179 (14)	113 (19)	
21–24	322 (48)	615 (49)	269 (46)	
25–30	135 (20)	298 (24)	117 (20)	

^a^ IQR refers to inter-quartile range. ^b^ Significance level determined that the non-parametric Kruskal-Wallis test was used for assessing continuous independent variables, and chi-square tests for the categorical variables. ^c^ Column percentages, because of rounding, may not be exact 100.

**Table 3 ijerph-15-02881-t003:** Multivariable associations of psychological well-being as a dichotomous variable (*N* = 2495).

Variables	Psychological Well-Being		
	Model 1	Model 2	Model 3
	AOR (95% CI)	AOR (95% CI)	AOR (95% CI)
Normal sleep quality	0.97 (0.92–1.003)		1.05 (1.09–1.23)
Poor sleep quality		1.27 (1.12–1.32)	1.26 (1.21–1.32)
Gender			
Female	Reference	Reference	Reference
Male	1.48 (1.04–1.95)	1.56 (1.10–2.06)	1.57 (1.10–2.07)
Education			
985 project	Reference	Reference	Reference
211 project	1.84 (1.12–2.84)	1.85 (1.13–2.85)	1.85 (1.13–2.86)
Yi Ben	1.60 (0.81–2.98)	1.69 (0.84–3.17)	1.81 (0.85–3.21)
Er Ben	1.60 (0.74–3.18)	1.50 (0.69–3.01)	1.54 (0.71–3.09)
San Ben	1.57 (0.75–3.20)	1.48 (0.66–2.98)	1.53 (0.69–3.01)
Household conditions			
State & society administrator (cadre in governments), Manager (Manager or department leader in state company, private company, foreign company and joint company)	Reference	Reference	Reference
Private enterprise owner (having large scale company), Technological specialist (teacher, engineer, doctor, academic researcher, lawyer, culture worker)	1.00 (0.48–1.93)	0.93 (0.44–1.79)	0.92 (0.44–1.78)
Clerk & office worker (secretary, accountant, computer operator), Individual entrepreneur (one-person company with unlimited liability)	1.25 (0.77–1.89)	1.23 (0.76–1.85)	1.22 (0.75–1.86)
Commercial & service worker (tertiary industry workers), Industrial worker (worker in factory, mining and transport industry)	1.11 (0.62–1.84)	1.05 (0.59–1.72)	1.06 (0.60–1.73)
Farmer, Unemployed	1.13 (0.60–2.01)	1.06 (0.61–1.77)	1.07 (0.63–1.82)
Marital or relationship status			
Yes	Reference	Reference	Reference
No	1.03 (0.69–1.35)	1.05 (0.72–1.37)	1.04 (0.70–1.36)
Physical activity			
Physically inactive	Reference	Reference	Reference
Less active (1–4 occasions per month)	3.31 (1.07–2.38)	3.28 (0.99–2.27)	3.39 (1.09–3.41)
More active (5+ occasions per month)	0.74 (0.46–1.21)	1.35 (0.76–1.32)	0.63 (0.39–1.19)
Body mass index	3.07 (1.21–2.66)	3.49 (1.38–2.79)	1.96 (1.06–2.13)
Place of birth			
Rural	Reference	Reference	Reference
Urban	0.77 (0.47–1.12)	0.75 (0.45–1.10)	0.76 (0.46–1.11)
Subjects			
Humanities and Social Sciences	Reference	Reference	Reference
Science	1.31 (0.96–1.48)	1.33 (1.02–1.51)	1.27 (0.93–1.32)
Engineering	1.34 (1.03–1.56)	1.29 (0.94–1.33)	1.36 (1.05–1.41)
Age			
18	Reference	Reference	Reference
19–20	0.76 (0.49–1.24)	0.77 (0.50–1.27)	0.78 (0.51–1.28)
21–24	0.85 (0.52–1.47)	0.88 (0.54–1.52)	0.87 (0.53–1.50)
25–30	1.03 (0.50–2.17)	1.15 (0.57–2.33)	1.09 (0.54–2.25)

**Table 4 ijerph-15-02881-t004:** Multivariable associations of psychological well-being operationalized in tertiles (*N* = 2495).

Variables	Mental Health		
	Model 1	Model 2	Model 3
	AOR (95% CI)	AOR (95% CI)	AOR (95% CI)
Normal sleep quality	0.96 (0.91–0.99)		1.03 (0.92–1.01)
Poor sleep quality		1.21 (1.13–1.68)	1.19 (1.09–1.57)
Gender			
Female	Reference	Reference	Reference
Male	1.71 (1.34–1.81)	1.75 (1.42–1.87)	1.79 (1.51–1.93)
Education			
985 project	Reference	Reference	Reference
211 project	1.62 (0.97–1.79)	1.50 (0.93–1.68)	1.64 (0.99–1.93)
Yi Ben	1.51 (0.94–1.69)	1.47 (0.86–1.53)	1.56 (0.97–1.71)
Er Ben	1.49 (0.89–1.57)	1.52 (0.95–1.65)	1.54 (0.96–1.67)
San Ben	1.42 (0.78–1.27)	1.45 (0.81–1.32)	1.48 (0.89–1.56)
Household conditions			
State & society administrator (cadre in governments), Manager (Manager or department leader in state company, private company, foreign company and joint company)	Reference	Reference	Reference
Private enterprise owner (having large scale company), Technological specialist (teacher, engineer, doctor, academic researcher, lawyer, culture worker)	1.36 (0.85–1.54)	1.31 (0.81–1.46)	1.29 (0.76–1.31)
Clerk & office worker (secretary, accountant, computer operator), Individual entrepreneur (one-person company with unlimited liability)	1.49 (0.93–1.72)	1.46 (0.91–1.69)	1.44 (0.89–1.63)
Commercial & service worker (tertiary industry workers), Industrial worker (worker in factory, mining and transport industry)	1.42 (0.87–1.59)	1.38 (0.86–1.55)	1.41 (0.87–1.51)
Farmer, Unemployed	1.45 (0.88–1.61)	1.32 (0.83–1.41)	1.39 (0.86–1.57)
Marital or relationship status			
Yes	Reference	Reference	Reference
No	0.92 (0.73–1.19)	0.92 (0.74–1.23)	0.92 (0.73–1.18)
Physical activity			
Physically inactive	Reference	Reference	Reference
Less active (1–4 occasions per month)	4.06 (1.49–2.88)	3.98 (1.21–2.65)	4.11 (1.53–2.92)
More active (5+ occasions per month)	1.05 (0.99–1.76)	1.57 (1.12–2.01)	0.83 (0.76–1.26)
Body mass index	3.12 (1.24–2.75)	3.57 (1.39–2.85)	2.35 (1.19–2.08)
Place of birth			
Rural	Reference	Reference	Reference
Urban	0.67 (0.79–1.03)	0.61 (0.71–0.94)	0.65 (0.75–1.01)
Subjects			
Humanities and Social Sciences	Reference	Reference	Reference
Science	1.46 (0.99–1.79)	1.51 (1.08–1.89)	1.37 (0.88–1.47)
Engineering	1.57 (1.15–1.90)	1.32 (0.79–1.38)	1.59 (1.17–1.94)
Age			
18	Reference	Reference	Reference
19–20	1.19 (0.77–1.61)	1.21 (0.78–1.63)	1.20 (0.77–1.62)
21–24	1.05 (0.63–1.50)	1.08 (0.66–1.54)	1.05 (0.63–1.52)
25–30	1.42 (0.76–2.41)	1.50 (0.79–2.54)	1.45 (0.76–2.46)

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
