# Peer review of "The Role of Sleep Quality in the Psychological Well-Being of Final Year UndergraduateStudents in China"

_ijerph, 2018, doi:10.3390/ijerph15122881_

Reviewer 1 Report

The title could be misleading – it could be mental health effecting sleep (such as anxiety), or poor sleep effecting mental health.

Abstract line 11 – “transition to society” is imprecise. They are already in society, but they will now be in the workplace.

The manuscript could use some minor English editing for clarity and flow. E.g. lines 39 – 40 “there are a variety of variations”

In the introduction lines 33 – 40 there appears to be a contradiction in statements. The authors state that university students have the same mental illness as the general population, but then go to state that they are more vulnerable and exhibit higher rates of mental illness.

Line 44 – poor sleep quality is not a disease, it is a symptom

Line 47 – English clarity

There introduction does not include enough specific information – there are generalities talked about, without exact numbers. In addition a discussion of how social pressures to perform in different countries is important. The Chinese context would likely be very different from typical Western education systems. More specifics should be given, and issues such as shared accommodation should be discussed in detail.

In the introduction there is irrelevant information such as the gender differences, or lack of them.

Methods:

Which university were the students from? Why is there not Chinese ethics approval, but only Glasgow?

Please explain the purpose of the random voucher value.

Please explain the K6 in more detail regarding how it relates to ICD or DSM diagnostic criteria.

Results:

Table 1 legend does not have enough information to interpret the results. What does the p-value represent? What are the comparators? This comment also applies to all other tables. There needs to be more information about what the tables and values mean in the table legend.

It would be interesting to know how the time of year and the time in relation to when exams are occurring effect mental health.

Some of the descriptions of results are worded quite right – some of them imply very large contributions and factors that aren’t correct.

It is unclear why multiple models are presented and how they were chosen.

Discussion:

Some discussion of how the various factors relate mechanistically would be useful. There is a vast and well documented literature on specific mechanisms of sleep disruption and mental health.

There are some limitations in the study that need to be discussed such as subjective measures, lack of ICD or DSM related outcomes, and lack of sleep duration metrics.

In addition, mental health is a strong term to use for this data given that clinical metrics are not used. Perhaps alternative wording such as psychological well-being would be more appropriate.

Author Response

Response to Reviewer 1 comments

First of all, we would like to give sincere thanks to the reviewer for taking the time and reviewing our paper. According to the valuable suggestions, we have revised this paper, which particularly helped to clarify some inaccuracies and restructure this paper. As a result, we believe that the quality of the paper has much improved. Please see below our response to the reviewer comments (held in red) and a description of the changes in the manuscript.

Point 1: The title could be misleading – it could be mental health effecting sleep (such as anxiety), or poor sleep effecting mental health.

Response: Yes, after reading the title, we also feel it could be misleading. We change the expression and make the title much clear. The new title is: The role of Sleep Quality in Mental Health of Last Year Undergraduate Students in China, which has been changed in manuscript. 

Point 2: Abstract line 11 – “transition to society” is imprecise. They are already in society, but they will now be in the workplace.

Response: Ok, thank you for this advice. We replace ‘transition to society’ with ‘transition to workplace’.

Point 3: The manuscript could use some minor English editing for clarity and flow. E.g. lines 39 – 40 “there are a variety of variations”

Response: We am sorry to bring misunderstanding due to the imprecise expression. Actually, here we mean factors. We change the word in manuscript already.  

Point 4: In the introduction lines 33 – 40 there appears to be a contradiction in statements. The authors state that university students have the same mental illness as the general population, but then go to state that they are more vulnerable and exhibit higher rates of mental illness.

Response: UK university students have the same mental illness as the general population, but the Hong Kong university students are more vulnerable and exhibit higher rates of mental illness, which is determined by different context. These sentences are not precisely expressed, so we rewrite them and provide more details. The different context determines the different research results, and thus studying university students’ sleeping quality and mental health should be closely related to specific context, which has also been discussed in terms of the reasons of choosing China as context.  

Point 5: Line 44 – poor sleep quality is not a disease, it is a symptom. 

Response: We are sorry for this typo. Yes, insomnia is disease, but poor sleep quality is not. We have replaced disease with symptom. Thank you for your advice.   

Point 6: Line 47 – English clarity

Response: We have made the sentence shorter and clarified the English expressions in Line 59 (revised version). 

Point 7: There introduction does not include enough specific information – there are generalities talked about, without exact numbers. In addition a discussion of how social pressures to perform in different countries is important. The Chinese context would likely be very different from typical Western education systems. More specifics should be given, and issues such as shared accommodation should be discussed in detail. In the introduction there is irrelevant information such as the gender differences, or lack of them.

Response: Thank you very much for this comment. In order to include enough specific information, we change some expressions in Introduction and add more details. We have also added details, including the importance of context (in line 41-42) and the reasons why we chose Chinas as the context (line 72-75; line 87-89). Some irrelevant information such as wrong expressions of gender differences has been deleted in order to keep consistent in whole paper. The rest of information of gender is kept in manuscript, because it can clarify the selection of variable of gender in statistical model. 

Point 8: Which university were the students from? Why is there not Chinese ethics approval, but only Glasgow?

Response: The university students were recruited across China’s various universities, including over 510 China’s universities. The paper is part of first author’s PhD working, so ethical approval was obtained in Glasgow, instead of Chinese ethics approval.  

Point 9: Please explain the purpose of the random voucher value.

Response: First, it can express thanks to participants, and additionally it could attract more participants and in some degree could increase the questionnaire completion quality. This information is missing in the previous version and it now has been added in revised manuscript. 

Point 10: Please explain the K6 in more detail regarding how it relates to ICD or DSM diagnostic criteria.

Response: We chose the K6 scale on mental well-being to measure the mental disorders, classified according to the Diagnostic and Statistical Manual of Mental Disorders, 4th Edition (DSM-IV). The K6 is the reliable and valid instrument to screen for any DSM-IV disorder [Cornelius et al., 2013], and additionally this measurement has been widely used to investigate the mental health of different population in China. Due to the reliability and validity of K6 , we chose it, and of course, the missing information has also been added in revised manuscript. 

Point 11: Results: Table 1 legend does not have enough information to interpret the results. What does the p-value represent? What are the comparators? This comment also applies to all other tables. There needs to be more information about what the tables and values mean in the table legend.

Response: We are sorry that we missed the notes. It should follow the tables. In revised manuscript, we added the missing notes (line 219-222). Following supplementary notes, the tables are understandable now. In the following tables, notes are added as well. 

Point 12: It would be interesting to know how the time of year and the time in relation to when exams are occurring effect mental health.

Response: Yes, we totally agree with you. Knowing the time is very interesting. Line 138 includes the time information.

Point 13: Some of the descriptions of results are worded quite right – some of them imply very large contributions and factors that aren’t correct. It is unclear why multiple models are presented and how they were chosen.

Response: We reviewed and revised all the descriptions of results, and corresponding changes are made. In terms of the reasons that we chose multivariable model, it was missed in manuscript. In the revised version, we added the missing information (see line 186-189). 

Point 14: Some discussion of how the various factors relate mechanistically would be useful. There is a vast and well documented literature on specific mechanisms of sleep disruption and mental health. There are some limitations in the study that need to be discussed such as subjective measures, lack of ICD or DSM related outcomes, and lack of sleep duration metrics.

Response: Thank you for this comment. We added the information about how the various factors relate mechanistically. It is useful when we talk about research results and discussion. The factors are also associated with relevant literature. With respect to limitations, we write it at the end of Conclusion section. We added the limitations you suggested. The suggestions are very conductive and valuable. We rewrite the limitations. 

Point 15: In addition, mental health is a strong term to use for this data given that clinical metrics are not used. Perhaps alternative wording such as psychological well-being would be more appropriate.

Response: Yes, after considering this comment, we found the term of mental health is very strong. We replace mental health with psychological well-being. It is more precise. 

Reviewer 2 Report

Thank you for the submission on “Effect of Sleep Quality on Mental Health of Last Year Undergraduate Students in China”

INTRODUCTION

Line 47 - … is a preventable problem??

Line 56-57 – provide references of the extant literature.

Line 57 – There are few researches? Please provide references of the few researches

Line 58-60 – Besides,… It is too convoluted

Line 61-62 – The sleep quality would be influenced by a variety of factors, such as… And please provide references

Line 62-65 – Most importantly… It is too convoluted.

Line 70-81 – This paper… Rewrite the paragraph

Although there is rich research investigating sleep and mental health of young adults (REFERENCES), few research explores the last year university students in China (REFERENCES). Besides, only a few research provides evidence showing how sleep quality affects mental health (REFERENCES), though there is plenty of research measuring mental health of young adults (REFERENCES).

Extant research has focused on mental health analysis of young adults (REFERENCES), but an increasing number of university students are facing more diverse mental problems, such as ... (REFERENCES).

Additionally, more and more factors are affecting university students’ mental health which include?? (provide also REFERENCES)

Why not use “final year” instead of “last year” Please change

English editing and proper referencing is required

METHODS

Is it possible to adjust also for regional or the university? Why could the researchers use a multilevel analysis because students are imbedded in universities which are also imbedded in different regions?

Line 157 – …using Stata 14, we analyzed all the data.

Line 160 – Why p < 0.20 and not any other?

Ethical concern: How did you anonymize the data bearing in mind that a voucher was created that may have require some personal details? What about the informed consent?

RESULTS

Line 167 – …categorical variables, respectively. Punctuation

Table 1 – I expected to see a 2x2 or RxC table by mental health problems or by sleep quality. What is the significance of a univariate Chi-square?

Table 2 – Why report mean (SD) and IQR instead of categorizing the sleep quality and giving n (%). Don’t you think it is more meaningful to categorize into “Low, Moderate and High”

How would you interpret, for instance, poor sleep quality in Tables 3 and 4? As the poor sleep quality score increases by one unit, the students are 27% more likely to have mental health. If the scores are categorized into different levels, then it would give a more meaningful interpretation.

In some cases, especially, in Table 2, you would not know whether you are dealing with n and row or column percentages or mean (SD).

For example, in Table 2, what is the percentage of males who have a high mental health as compared to females? Row percentages of each category is valuable. In Table 1 we already know that there are 47% females, why repeat that in Table 2? What then is the reason for Table one of univariate Chi-square? Please merge Tables 1 and 2.

I am a biostatistician and an epidemiologist, but I had to struggle to read your tables. I had to give it a second look just to realize that you were using raw scores for the regression analysis of sleep quality indicators. Your tables are not clearly understood. Revise accordingly and present meaningful results.

 DISCUSSION

Line 216 – This study of “last year” undergraduate students in China… Using “last year” may mean a different thing. Last year may be assumed to be 2017 if we are in 20 18. Please use “final year.”

Line 227 – The referencing of Markus et al. 2014. Are you using numbering format?

English editing, sentence structure and proper referencing is required.

 Author Response

Response to Reviewer 2 comments

First of all, we would like to give sincere thanks to the reviewer for taking the time and reviewing our paper. According to the valuable suggestions, we have revised this paper, which particularly helped to clarify some inaccuracies and restructure this paper. As a result, we believe that the quality of the paper has much improved. Please see below our response to the reviewer comments (held in red) and a description of the changes in the manuscript.

Point 1: Line 47 - … is a preventable problem??

Response: We are very sorry that it is a typo. It was corrected by Word’s automatic error correction. And the right word should be prevalent. It is a prevalent problem, which has been corrected. 

Point 2: Line 56-57 – provide references of the extant literature. Line 57 – There are few researches? Please provide references of the few researches

Response: We missed the references, and in the revised version, we added them in line 75.

Point 3: Line 58-60 – Besides,… It is too convoluted.

Response: It is too long, so we change the convoluted sentence into two simple sentences (see line 76).

Point 4: Line 61-62 – The sleep quality would be influenced by a variety of factors, such as… And please provide references

Response: Here, it actually is expressed wrongly. The sentence follows the China’s common accommodation issue. We want to say the situation in China is affected by a variety of factors. Corrections are made in line 77-81. 

Point 5: Line 62-65 – Most importantly… It is too convoluted.

Response: We make the long sentences into several simple sentences. In addition, we move the sentence (More importantly……sleep problems) to line 77, because it is related to the last sentence (Besides, last year university students in China are a special group, because they are undergoing a critical transitory period in which they are going from university life to adulthood and making major life decisions [5]). Now it is easily understood. 

Point 6: Line 70-81 – This paper… Rewrite the paragraph. Although there is rich research investigating sleep and mental health of young adults (REFERENCES), few research explores the last year university students in China (REFERENCES). Besides, only a few research provides evidence showing how sleep quality affects mental health (REFERENCES), though there is plenty of research measuring mental health of young adults (REFERENCES). Extant research has focused on mental health analysis of young adults (REFERENCES), but an increasing number of university students are facing more diverse mental problems, such as ... (REFERENCES). Point 9: Additionally, more and more factors are affecting university students’ mental health which include?? (provide also REFERENCES)

Response: This paragraph is not well supported by references, and it is not cohesive. We rewrite it. Please see line 91-99. Some repetitive and subjective sentences are deleted, for they have been discussed in the above paragraphs.

Point 7: Why not use “final year” instead of “last year” Please change

Response: Fine. Final year is more academic and formal. Thank you very much for this suggestion. We replace all last year with final year.

Point 8: English editing and proper referencing is required.

Response: Proofreading has been done by a native speaker expertizing in this research field.

Point 9: Is it possible to adjust also for regional or the university? Why could the researchers use a multilevel analysis because students are imbedded in universities which are also imbedded in different regions?

Response: The reasons that we use multivariable model have been provided in line 502-507.

Point 10: Line 157 – …using Stata 14, we analyzed all the data.

Response: We adopt this correct sentence. And the wrong sentence has been deleted. 

Point 11: Line 160 – Why p < 0.20 and not any other?

Response:extant research shows a p<0.20 can avoid the overcontrolling (Primack et al., 2018). And thus the study analyzed data by using covariates with a p < 0.20

Point 12: Ethical concern: How did you anonymize the data bearing in mind that a voucher was created that may have require some personal details? What about the informed consent?

Response: Because the voucher was send out by Wechat software, the data was still anonymized. Collecting voucher in Wechat does not need participants’ personal information.

Point 13: Line 167 – …categorical variables, respectively. Punctuation

Response: The missing punctuation is added. Thank you for your detailed feedback. 

Point 14: Table 1 – I expected to see a 2x2 or RxC table by mental health problems or by sleep quality. What is the significance of a univariate Chi-square?

Response: Thank you so much for this comment. We marked the 2x2 table in red (in table 1 and 2). Because there is plenty of sociodemographic variables, we put all variables together. If we split tables into many ones, it will become in chaos. In addition, we are sorry that we missed the notes at the end of table 1 and 2. In the revised manuscript, we add the missing information and explain why we use Chi-square. 

Point 15: Table 2 – Why report mean (SD) and IQR instead of categorizing the sleep quality and giving n (%). Don’t you think it is more meaningful to categorize into “Low, Moderate and High”

Response: In multivariable models we use in this paper, SD and IQR are two important indicators. IQR refers to inter-quartile range, and it can show the concentrated trend and can check the exceptional value. Because sleep quality is studied, it is very hard to assess sleep quality through subjective report. If we set three options (low, moderate and high), participants will feel hard to choose. In this situation, participants tend to choose moderate, which is found in some similar research. Thus, we only use two options of positive and negative. This is explained in line 463-466.

Point 16: How would you interpret, for instance, poor sleep quality in Tables 3 and 4? As the poor sleep quality score increases by one unit, the students are 27% more likely to have mental health. If the scores are categorized into different levels, then it would give a more meaningful interpretation.

Response: The poor sleep quality was assessed by participants. We got the questionnaire results. In the models, we did not provide sub-categories of poor sleep quality. Yes, as the poor sleep quality score increases by one unit, the students are 27% more likely to have problems in psychological well-being. It is a limitation in this paper. According to participant’s feedback on questionnaire, it is very hard to give categorized sleep quality into many levels.  

Point 17: In some cases, especially, in Table 2, you would not know whether you are dealing with n and row or column percentages or mean (SD).

Response: Taking Table 2 as an example, we presented the whole sample characteristics. Because the notes were missing, it was very hard to follow. In revised version, we added the missing notes at the end of tables, and thus it is understandable now.

Point 18: For example, in Table 2, what is the percentage of males who have a high mental health as compared to females? Row percentages of each category is valuable. In Table 1 we already know that there are 47% females, why repeat that in Table 2? What then is the reason for Table one of univariate Chi-square? Please merge Tables 1 and 2.

Response: The purpose of conducing Chi-square was missing, but now we add the notes. In line 526-527, we also explained. In terms of the repetition, we delete the repetitive contents in Table 2 in the revised version. The Table 1 shows the descriptive statistics while the Table 2 includes the bivariables associations, and actually there are two different tables, so we did not merge them. Additionally, ff we merge them, the table would be too large. 

Point 19: I am a biostatistician and an epidemiologist, but I had to struggle to read your tables. I had to give it a second look just to realize that you were using raw scores for the regression analysis of sleep quality indicators. Your tables are not clearly understood. Revise accordingly and present meaningful results.

Response: Yes, you are right. After we consider your comments, we merge repetitive contents. More importantly, we add the missing notes. Based on the notes and our revision, the tables are now understandable.  

Point 20: Line 216 – This study of “last year” undergraduate students in China… Using “last year” may mean a different thing. Last year may be assumed to be 2017 if we are in 20 18. Please use “final year.”

Response: Yes. Thank you for this feedback. It brings misunderstandings. We replace the final year with final year. 

Point 21: Line 227 – The referencing of Markus et al. 2014. Are you using numbering format?

Response: The reference format was wrong. In revised version, we use the correct format. 

Point 22: English editing, sentence structure and proper referencing is required.

Response: Yes. Proofreading has been done by a native speaker expertizing in this research field. 

 Round  2

Reviewer 1 Report

Thanks for the good responses to comments and changes. I am happy with the changes.

Only 1 minor correction, the word "textiles" is used when I think "tertiles" or another word should be used.

Author Response

Thank you very much for your time and concern. We are glad that our revision can meet your standard. 

Yes, tertiles should be used, instead of textiles. Because of WORD's automatic error correction, it becomes typos. We are sorry about it. We have changed all wrong words. 

Thank you very much again.

Reviewer 2 Report

Please check on minor errors and typos such as:

"...increase in poor sleep quality would resulted in a 26% increase.."

Author Response

Thank you for your time and concern. According to your feedback, we revise the manuscript again. And we mainly concentrate on typos. The sentence in your review report has been corrected.